# Volume-Preserving Shear Transformation of an Elliptical Slant Cone to a Right Cone

Marco Frego [1,*,†] and Cristian Consonni [2,†,‡]

[1] Faculty of Engineering, Free University of Bozen-Bolzano, Via Volta 13/A, 39100 Bolzano, BZ, Italy
[2] Joint Research Centre—European Commission, Via Enrico Fermi, 2749, 21027 Ispra, VA, Italy; cristian.consonni@ec.europa.eu
[*] Correspondence: marco.frego@unibz.it
[†] The authors contributed equally to this work.
[‡] Work Done While at Eurecat—Centre Tecnològic de Catalunya, Unit of Big Data and Data Science, 08005 Barcelona, Spain.

**Abstract:** One nappe of a right circular cone, cut by a transverse plane, splits the cone into an infinite frustum and a cone with an elliptical section of finite volume. There is a standard way of computing this finite volume, which involves finding the parameters of the so-called shadow ellipse, the characteristics of the oblique ellipse (the cut) and, finally, the projection of the vertex of the cone onto the oblique ellipse. This paper shows that it is possible to compute that volume just by using the information of the shadow ellipse and the height of the cone. Indeed, the finite slant cone has the same volume of an elliptic right cone, with the base being the shadow ellipse of the cut portion and with the height being the distance between the vertex of the cone and the intersection of the height of the original cone with the cutting plane. This is proved by introducing a volume-preserving shear transformation of the elliptical slant cone to a right cone, so that the standard volume formula for a cone can be straightforwardly applied. This implies a simplification in the procedure for computing the volume, since the oblique ellipse—i.e., the difficult part—can be neglected because only the shadow ellipse needs to be determined.

**Keywords:** cone; frustum; volume; conic section; shadow ellipse

**MSC:** 26B15; 28-01; 28A75; 51M04; 51M25; 97G30; 97G40

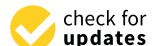



## 1. Introduction and Problem Formulation

This paper presents a simplified—and novel, to the best of the authors' knowledge—method for computing the volume of the finite portion obtained by cutting one nappe of a right circular cone with a transverse plane, so that the cut section is elliptical; see Figure 1.

Without loss of generality, it is possible to recast the problem in standard form: the nappe of the circular right cone $C$ is assumed to have a Cartesian equation of the form (see Figure 2 for reference to the quantities introduced next):

$$C: \ z = \sqrt{c^2(x^2 + y^2)},$$

where $c > 0$ defines the opening of the cone from its vertex $V$ and it is related to the half-angle $\alpha$ by the formula

$$c = \tan(\pi/2 - \alpha) = \cot(\alpha).$$

A transverse cutting plane $P$ has Cartesian coordinates, for real coefficients $a_1, \ldots, a_4$,

$$P: \ a_1 x + a_2 y + a_3 z + a_4 = 0,$$

but, since only a finite volume is considered, parabolic and hyperbolic sections are excluded. Moreover, by symmetry, it is always possible to rotate around the $z$ axis, so that the equation of the plane $P$ reduces to $z = mx + q$, with nonnegative $m$ and $q$. The plane cuts the cone with an angle $\varphi$ obtained as $\tan \varphi = m$. These conditions imply that $0 < m < c$, or equivalently, that $0 < \varphi < \pi/2 - \alpha$ (inequalities are strict to avoid a degenerate cut).

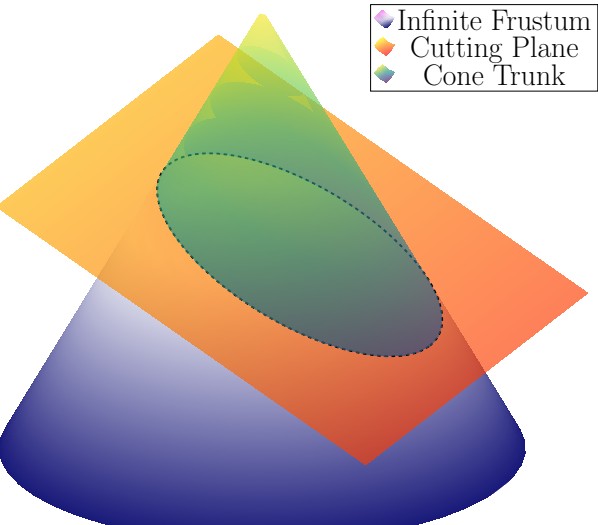

**Figure 1.** A nappe of a right circular cone cut by a transverse plane divides the nappe into a finite volume trunk and an infinite frustum.

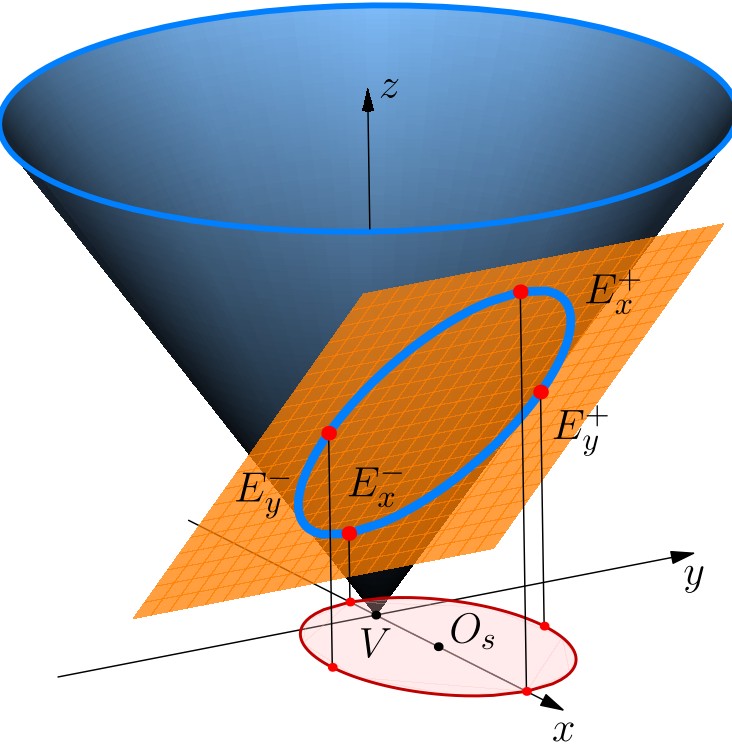

**Figure 2.** *Cont.*

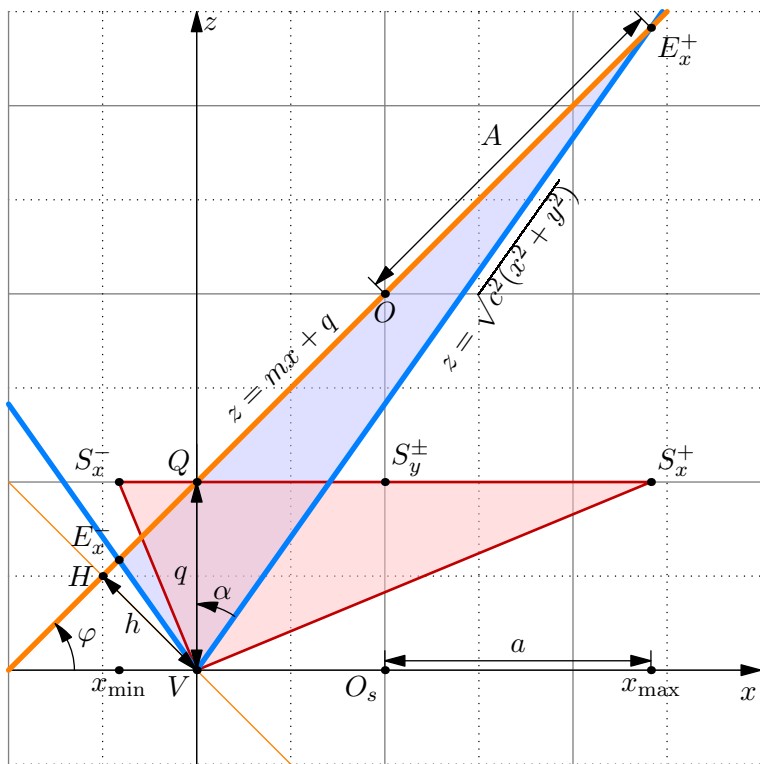

**Figure 2.** (**Top**) The right circular blue cone is cut by a transverse orange plane: the cut section is elliptical and the four vertices of the oblique ellipse are identified by the points $E_x^+$, $E_x^-$, $E_y^+$, and $E_y^-$. The shadow ellipse (in red) is characterised by the projection of the oblique ellipse onto the plane $z = 0$. $V$ is the vertex of the cone, $O_s$ is the centre of the shadow ellipse. (**Bottom**) Section view of the cut cone. The finite volume slant cone to be found is shaded in blue, the shear-transformed right elliptical cone with the same volume is shaded in red.

## 2. Background and State of the Art

Before illustrating our new procedure to compute the volume of the cut cone using a shear transformation, we revise the usual way, presented in [1], i.e., to integrate slices of the cone parallel to the cutting plane along the height $VH$ of the cone, as in Figure 3 (left), which needs the following steps to be performed. It is necessary to find the parameters of the oblique ellipse and the projection $H$ of the vertex $V$ on the plane $P$. The first step for this computation is to find the *shadow ellipse*, that is, the ellipse obtained intersecting the cone $C$ with the plane $P$, namely

$$\sqrt{c^2(x^2 + y^2)} = mx + q, \text{ that is}$$

$$y = \pm \frac{\sqrt{(m^2 - c^2)x^2 + 2mqx + q^2}}{c} \text{ or}$$

$$x = \frac{mq \pm \sqrt{c^2 y^2(m^2 - c^2) + c^2 q^2}}{c^2 - m^2}.$$

The above implicit equations with respect to $x$ or $y$ represent the shadow ellipse that lays in the plane $z = 0$, with axes parallel to $x$ and $y$, which is depicted in red in Figure 2. Notice that the aforementioned condition $0 < m < c$ ensures that the denominator $c^2 - m^2$ never vanishes.

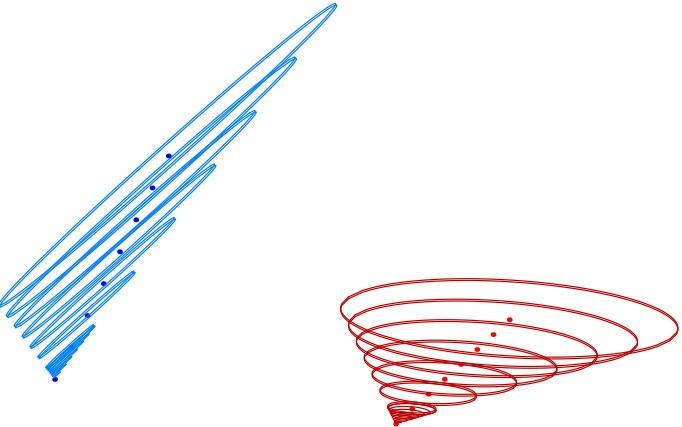

**Figure 3.** The **left** cone is sliced in the usual way: parallel to the cutting plane. The **right** cone has the same volume and has the basis parallel to $z = 0$. The dots are the centres of the ellipses.

To find the parameters of this ellipse, it is possible to use the polar coordinates or to study the interval of existence of the above radicals, in both cases, the result is $x \in [x_{\min}, x_{\max}], y \in [y_{\min}, y_{\max}]$, where

$$
\begin{aligned}
x_{\min} &= -\frac{q}{c+m}, & x_{\max} &= \frac{q}{c-m}, \\
y_{\min} &= -\frac{q}{\sqrt{c^2-m^2}}, & y_{\max} &= \frac{q}{\sqrt{c^2-m^2}}.
\end{aligned}
\tag{1}
$$

Thereby, the major semiaxis $a$, the minor semiaxis $b$ and the centre of the shadow ellipse $O_s$ are

$$
a = \frac{cq}{c^2-m^2}, \quad b = \frac{q}{\sqrt{c^2-m^2}}, \quad O_s = \left(\frac{mq}{c^2-m^2}, 0\right) \equiv (g, 0).
\tag{2}
$$

The oblique ellipse in space, corresponding to the intersection of $C$ and $P$, is the image of the shadow ellipse mapped on the cone or on the plane. It has vertices (see Figure 2),

$$
\begin{aligned}
E_x^- &= \left(x_{\min}, 0, \frac{cq}{c+m}\right), & E_x^+ &= \left(x_{\max}, 0, \frac{cq}{c-m}\right), \\
E_y^- &= \left(g, y_{\min}, \frac{c^2q}{c^2-m^2}\right), & E_y^+ &= \left(g, y_{\max}, \frac{c^2q}{c^2-m^2}\right),
\end{aligned}
\tag{3}
$$

where $g \equiv \frac{mq}{c^2-m^2}$, as in (2) and the other values are given in (1). They define the oblique ellipse, as in Figure 3, which has semiaxes $A$, $B$ and centre $O$ given by

$$
\begin{aligned}
A &= \frac{1}{2}\mathrm{dist}(E_x^+, E_x^-) = \frac{cq\sqrt{m^2+1}}{c^2-m^2}, \\
B &= \frac{1}{2}\mathrm{dist}(E_y^+, E_y^-) = \frac{q}{\sqrt{c^2-m^2}}, \\
O &= \left(g, 0, \frac{qc^2}{c^2-m^2}\right).
\end{aligned}
\tag{4}
$$

To compute the volume using the standard method, it is necessary to know the height of the cone, which corresponds to the length $h$ in Figure 2. This is the distance from the projection of point $V$ onto plane $P$, or equivalently, the length of segment $VH$, where

$$H = \left( -\frac{qm}{m^2+1},\ 0,\ \frac{q}{m^2+1} \right),$$

$$\text{dist}(V, H) = h = \frac{q}{\sqrt{m^2+1}}. \tag{5}$$

In conclusion, the volume of the cone is the area of the base, i.e., the oblique ellipse (4), times a third of the height $h$:

$$\text{Vol} = \frac{\pi}{3} ABh = \frac{\pi}{3} \frac{cq^3}{(c^2-m^2)^{\frac{3}{2}}}. \tag{6}$$

The next section will show that the above volume can be computed by solely considering the shadow ellipse (2), hence avoiding the computation of the parameters of the oblique space ellipse (4), and of the projection of $V$ onto $P$ (5).

### 3. Equivalence of the Volume of the Slanted and Right Cone

It is straightforward to verify that the cone with the base being the shadow ellipse and the height $VQ = q$, i.e., the intersection of the height of the original cone with the cutting plane, has a volume given by $\text{Vol} = \pi/3\,abq$, which is the value obtained in (6), after the substitution of (2):

$$
\begin{aligned}
\text{Vol} &= \int_0^q \pi a(z) b(z)\ \mathrm{d}z \\
&= \pi \int_0^q \frac{cz}{c^2-m^2} \cdot \frac{z}{\sqrt{c^2-m^2}}\ \mathrm{d}z \\
&= \frac{\pi c}{(c^2-m^2)^{\frac{3}{2}}} \int_0^q z^2\ \mathrm{d}z = \frac{\pi}{3} abq \\
&= \frac{\pi}{3} \frac{cq^3}{(c^2-m^2)^{\frac{3}{2}}},
\end{aligned}
$$

where $a(z)$ and $b(z)$ are the expressions (given in (2)) of the semi-axes of the ellipse integrated from 0 to $q$. Indeed, the usual way of computing the volume is to apply Cavalieri's Principle to the oblique cone, that is, to employ a *shear transformation*, ref. [2] (Ch. 5) which is a linear mapping that preserves volumes, as in Figure 3. We can map the slanted cone (left) to the right cone (right) slice by slice [3] (Ch. 4). For the slanted cone, if we intercept the nappe with the plane $z = mx + q'$ with $0 \le q' \le q$, then we obtain the analogous vertices of the oblique ellipse, as in (3), but with $q$ replaced by $q'$. Analogously, for the slices of the right cone, we obtain (2) with $q$ replaced by the $q'$ corresponding to each slice. If we compute the area $\texttt{Area}(E_{q'})$ of each slice of the slanted cone, we obtain

$$\texttt{Area}(E_{q'}) = \pi A_{q'} B_{q'} = \frac{\pi c q'^2 \sqrt{m^2+1}}{(c^2-m^2)^{\frac{3}{2}}},$$

while the area of each slice of the right cone is

$$\texttt{Area}(O_{q'}) = \pi a_{q'} b_{q'} = \frac{\pi c q'^2}{(c^2-m^2)^{\frac{3}{2}}}.$$

In other words, the ratio of the area of each slice of the slanted cone to the area of each slice of the right cone is

$$\frac{\texttt{Area}(E_{q'})}{\texttt{Area}(O_{q'})} = \sqrt{m^2+1}.$$

On the other hand, the height of the slanted cone is shorter than the height of the right cone by a factor $h/q = \frac{1}{\sqrt{m^2+1}}$. We can therefore prove that the two volumes are equal.

**Theorem 1.** *The finite volume of one nappe of a right circular cone of equation $z = \sqrt{c^2(x^2+y^2)}$ (for $c > 0$) cut by a transverse plane of equation $z = mx + q$ is equal to the volume of a cone with a base that is the shadow ellipse of semiaxes a and b (given in (2)) and a height equal to the distance $QV = q$, i.e., the intersection of the height of the original cone C with the cutting plane (Figure 2). Finally, the volume is*

$$\text{Vol} = \frac{\pi}{3} \frac{cq^3}{(c^2 - m^2)^{\frac{3}{2}}}. \tag{7}$$

**Proof.** The idea of the present simplification is to prove the existence of a shear transformation that maps the oblique ellipse to the shadow ellipse, so that some steps of the usual computation are spared, because the standard cone formula can be directly applied.

This corresponds to finding a change of coordinates in the integral for the volume that has a determinant of the Jacobian equal to 1. An elementary shear transform matrix $\boldsymbol{T}$ is used to map the oblique plane $P$ onto the horizontal plane $z = q$. This transformation is defined by a single parameter $k$, which needs to be determined, and is of the form

$$\boldsymbol{T} = \begin{pmatrix} 1 & 0 & 0 \\ 0 & 1 & 0 \\ k & 0 & 1 \end{pmatrix}. \tag{8}$$

A generic point $R$ on the cutting plane $P$ has coordinates $R = (x, y, mx + q)$. The application of $\boldsymbol{T}$ to the generic point $R$ yields a new point $R'$ such that

$$R' = \boldsymbol{T} \cdot R = \boldsymbol{T} \cdot \begin{pmatrix} x \\ y \\ mx + q \end{pmatrix} = \begin{pmatrix} x \\ y \\ mx + kx + q \end{pmatrix}.$$

Constraining $R'$ to lie on the plane $z = q$, we obtain the equation on the $z$-component: $(m + k)x + q = q$, from which we obtain $k = -m$.

Note that the transformation $T$ in (8) maps each point $P$ that belongs to a slice of the slanted cone onto a corresponding point $P'$ belonging to a slice of the right cone at height $q'$:

$$P' = \boldsymbol{T} \cdot P = \boldsymbol{T} \cdot \begin{pmatrix} x \\ y \\ mx + q' \end{pmatrix}$$

$$= \begin{pmatrix} x \\ y \\ mx - mx + q' \end{pmatrix} = \begin{pmatrix} x \\ y \\ q' \end{pmatrix}.$$

Applying $\boldsymbol{T}$ to the four vertices $E_x^+, E_y^-, E_y^+, and E_y^-$, we retrieve the vertices of the shadow ellipse on $z = q$ (see for reference Figure 2):

$$S_x^- = (x_{\min}, 0, q), \qquad S_y^- = (g, y_{\min}, q),$$

$$S_x^+ = (x_{\max}, 0, q), \qquad S_y^+ = (g, y_{\max}, q).$$

The slant cone is thus mapped to a right cone and its volume can be easily retrieved with the standard formula: area of the base times the height divided by three, as in (7). Indeed, the shear transformation $T$ has a unit determinant, which preserves volumes in three-dimensional spaces and areas in two-dimensional spaces. $\square$

## 4. Conclusions

This paper demonstrates that the volume of a slanted cone, which results from slicing one nappe of a right circular cone with a transverse plane, is equivalent to the volume of a right cone. This right cone has an elliptical base, which corresponds to the projection (or "shadow") of the slant cone onto the plane, and a height determined by the intersection of the original cone's height and the cutting plane. The function that maps the slant cone to the right cone is a shear transformation with a unit determinant, which preserves volumes in three-dimensional spaces and areas in two-dimensional spaces. Computing the cone's volume is thus simplified, since the oblique ellipse, and the vertex's projection on the cutting plane can be neglected because only the shadow ellipse must be determined and integrated over. This is equivalent to saying that only the semiaxes *a* and *b* (computed in (2)) are required: the computation of the oblique ellipse (4) and of the projection of the vertex (5) can be avoided.

Furthermore, this method can be used to compute the volume of any frustum generated by intersecting the cone with two planes.

**Author Contributions:** Conceptualization, M.F. and C.C.; Methodology, M.F. and C.C. The authors contributed equally to this work. All authors have read and agreed to the published version of the manuscript.

**Funding:** This research received no external funding.

**Informed Consent Statement:** Not applicable.

**Data Availability Statement:** Not applicable.

**Conflicts of Interest:** The authors declare no conflicts of interest.

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
