# Peer review of "Volume-Preserving Shear Transformation of an Elliptical Slant Cone to a Right Cone"

_axioms, doi:10.3390/axioms13040245_

Round 1

Reviewer 1 Report

Comments and Suggestions for Authors

In this paper, the authors compute the volume of the finite part of a cone after an elliptical section was done on this cone. So in this paper is proved that it is possible to compute that volume just by using the information of the  shadow ellipse and the height of cone. The authors remarked that the finite slant cone has the same volume of an elliptic right cone with base the shadow ellipse of the cut portion and with height the distance between the vertex of the cone and the  intersection of the height of the original cone with the cutting plane.

The proof of this important result was done by the authors using shear transformations of the elliptical slant cone to a right  cone. 

The paper is interesting, contains new results and present a way how isometries--- in this case shear transformation could be used for computations of some important results in the theory of quadrics.

In conclusion, the computation of  the cone’s volume is thus simplified, since the oblique ellipse, and the vertex’s projection on the cutting plane can be neglected because  solely the shadow ellipse must be determined and integrated over, as the authors remarked.

This new method presented by the authors could be also used in future computations for other type of quadrics and represent a new way to compute the volume of the cone using shadow ellipse. 

However, some typos are along the paper and I recommend to the authors to check carefully all the paper in this respect. For example, already at the begining of Section 1, is wroted "Thipresents paper" -- I think correct is "The present paper..."

After all the paper is checked carefully in this respect and the English of the paper is improved, I recommend for publication this paper.

Comments on the Quality of English Language

However, some typos are along the paper and I recommend to the authors to check carefully all the paper in this respect. For example, already at the begining of Section 1, is wroted "Thipresents paper" -- I think correct is "The present paper..."

Author Response

We would like to thank the Reviewer for the very useful and constructive comments. We performed a thorough revision of the paper and improved the readability thanks to the Reviewer's comments.

Reviewer 2 Report

Comments and Suggestions for Authors

There are many publications which are devoted to  volume –preserving maps. In the present paper,  authors  consider  a nappe of a right circular cone  cutting  by a transverse plane divides the nappe into a finite volume trunk (an elliptical slant cone) and an infinite frustum. They introduce the shadow ellipse  and a  new  volume-preserving shear  transformation  of the elliptical slant cone to a right cone. It is shown that the volume of the slanted cone is equal to the volume of the right cone that has  basis the shadow ellipse and height the intersection of the height of the original cone with  the cutting plane. 

 As consequence, the volume of the elliptical slant conecan be  easily  calculated. All figures  in the text have an  excellent quality.

My opinion: the

Paper is  suitable  for the journal Axioms

Author Response

(The authors gave the same response as above.)

Reviewer 3 Report

Comments and Suggestions for Authors

The referee report can be found in the attachment.

Comments on the Quality of English Language

Moderate editing of the English language is required.

Author Response

We would like to thank the Reviewer for the very useful and constructive comments. We performed a thorough revision of the paper and improved the readability thanks to the Reviewer's comments.

Regarding the suggestion for additional references, we have added a citation to a recent work (Miklavcic, 2020) that provides additional context to our work.